# Larvicidal Activities against *Aedes aegypti* of Supernatant and Pellet Fractions from Cultured *Bacillus* spp. Isolated from Amazonian Microenvironments

**DOI:** 10.3390/tropicalmed6020104

**Published:** 2021-06-17

**Authors:** Ricardo M. Katak, Elerson M. Rocha, Juan C. Oliveira, Veranilce A. Muniz, Marta R. Oliveira, Francisco A. S. Ferreira, William R. Silva, Rosemary A. Roque, Antonia Q. L. de Souza, Jayme A. Souza-Neto, Olle Terenius, Osvaldo Marinotti, Wanderli P. Tadei

**Affiliations:** 1Programa Multi-Institucional de Pós-Graduação em Biotecnologia—PPGBIOTEC, Universidade Federal do Amazonas—UFAM, Manaus 69042-270, Brazil; elerson.matos@hotmail.com (E.M.R.); juabio1807@gmail.com (J.C.O.); 2Programa de Pós-Graduação em Biotecnologia e Recursos Naturais da Amazônia—PPGMBT, Universidade do Estado do Amazonas—UEA, Manaus 69042-270, Brazil; veralves2@gmail.com; 3Programa de Pós-Graduação em Biodiversidade e Biotecnologia—PPGBIONORTE, Universidade Federal do Amazonas—UFAM, Manaus 69042-270, Brazil; moliveirabiotec@gmail.com; 4Programa de Pós-Graduação em Ciências Biológicas (Entomologia)—PPGENT, Instituto Nacional de Pesquisas da Amazônia—INPA, Manaus 69042-270, Brazil; fcoaugusto.bio@gmail.com (F.A.S.F.); wrds021@gmail.com (W.R.S.); 5Laboratório de Controle Biológico e Biotecnologia da Malária e Dengue—LCBBMD, Instituto Nacional de Pesquisas da Amazônia—INPA, Manaus 69042-270, Brazil; rosebio1996@yahoo.com.br (R.A.R.); wptadei@gmail.com (W.P.T.); 6Faculdade de Ciências Agrárias—FCA, Universidade Federal do Amazonas—UFAM, Manaus 69042-270, Brazil; antoniaqlsouza@gmail.com; 7School of Agricultural Sciences, Department of Bioprocesses and Biotechnology, Central Multi User Laboratory, São Paulo State University—UNESP, Botucatu 18600-000, Brazil; jayme.souza-neto@unesp.br; 8Department of Cell and Molecular Biology, Microbiology, Uppsala University, 74174 Uppsala, Sweden; olle.terenius@icm.uu.se; 9MTEKPrime, Aliso Viejo, CA 92656, USA

**Keywords:** Amazonian microbiota, bioprospecting, biological control, *Bacillus*, mosquito

## Abstract

The *Aedes aegypti* mosquito is the primary vector of Dengue, Chikungunya and Zika causing major problems for public health, which requires new strategies for its control, like the use of entomopathogenic microorganisms. In this study, bacteria from various Amazonian environments were isolated and tested for their pathogenicity to *A. aegypti* larvae. Following thermal shock to select sporulated *Bacillus* spp., 77 bacterial strains were isolated. Molecular identification per 16S RNA sequences revealed that the assembled strains contained several species of the genus *Bacillus* and one species each of *Brevibacillus*, *Klebsiella*, *Serratia*, *Achromobacter* and *Brevundimonas*. Among the isolated *Bacillus* sp. strains, 19 showed larvicidal activity against *A. aegypti*. Two strains of *Brevibacillus halotolerans* also displayed larvicidal activity. For the first time, larvicidal activity against *A. aegypti* was identified for a strain of *Brevibacillus halotolerans*. Supernatant and pellet fractions of bacterial cultures were tested separately for larvicidal activities. Eight strains contained isolated fractions resulting in at least 50% mortality when tested at a concentration of 5 mg/mL. Further studies are needed to characterize the active larvicidal metabolites produced by these microorganisms and define their mechanisms of action.

## 1. Introduction

Vector-borne diseases exert a huge burden of morbidity and mortality worldwide, particularly affecting the poor living in the tropics and subtropics. *Aedes* (*Stegomyia*) *aegypti* (L.) is considered the primary vector of arboviruses causing dengue, Zika and chikungunya illnesses [1,2,3]. The adaptability of these mosquitoes to urban areas, their highly anthropophilic behavior and recent outbreaks of the diseases mentioned above, make this vector an important public health problem to be solved [4,5,6]. Across the globe, dengue, Zika, and chikungunya outbreaks vary in geographic location, magnitude and duration. Between 95 and 114 million dengue infections are estimated to occur globally per year, with 32 to 66 million febrile disease cases, and 4 million infections that require hospitalization [7]. In 2015, a Zika outbreak in Brazil accounted for 440 thousand to 1.3 million infection cases, quickly spreading throughout Latin America and the Caribbean [8]. Zika was declared a public health emergency of international concern by the World Health Organization in February 2016 [9]. In the early 2000s, chikungunya outbreaks were reported in countries in Africa, Asia, and Europe. In 2013, it emerged in the Americas where within the first year, over a million new cases were reported [10].

The lack of preventive drugs for dengue, Zika and chikungunya, makes the mosquito vector the main target for the comprehensive control of these diseases [11]. Currently, the focus is on reducing the source of these mosquitoes, using mainly insecticides. However, the increased resistance of mosquitoes to chemical insecticides makes them nonfunctional in many environments [12,13]. New tools and approaches are needed to control the populations of *A. aegypti*.

Entomopathogenic microorganisms are promising tools for the control of disease-carrying mosquitoes [14,15,16,17,18]. Bacteria of the genus *Bacillus*, mainly *Bacillus thuringiensis israelensis* (Bti) and *Lysinibacillus sphaericus* (Syn. *Bacillus sphaericus*, Bs), have been used worldwide and for decades as biological products for the control of mosquitoes by killing their larvae [19,20,21]. Despite products based on Bti and Bs being powerful and selective insecticides, alternative *Bacillus* sp. strains should be sought as alternatives, potentially revealing new insecticide toxins.

Besides Bti and Bs, microorganisms such as the bacteria *Chromobacterium* sp. Panama [22], *Xenorhabdus* and *Photorhabdus* [23] and the fungi *Beauveria bassiana* and *Metarhizium anisopliae* among others, have demonstrated mosquitocidal activities [24]. The present challenge is to convert these promising observations into products that are ready to be incorporated into mosquito control interventions.

The Amazon tropical rainforest contains ~25% of the world’s terrestrial biodiversity [25], including insects, mammals, birds, and microorganisms. Here, we describe the isolation and characterization of bacteria from diverse Amazonian environments and their potential as new components of *A. aegypti* control programs. We identified bacterial strains that exhibit larvicidal activity similar to commercial Bti. Further studies are needed for the characterization of their active compounds. Metabolites derived from the isolated bacteria may be molecules with novel insecticidal properties that can be developed as biological tools to improve the control of mosquito-transmitted diseases.

## 2. Materials and Methods

### 2.1. Sample Collection and Bacteria Isolation

Water and soil samples were collected in the municipalities of Coari, Manaus and Parintins, Amazonas State, Brazil (Table 1, Figure 1). All the biological material collected and analyzed during the present study was collected with official permission (21263-1) given by ‘Sistema de Autorização e Informação em Biodiversidade’ (SISBIO) of the Brazilian Ministry of Environment (MMA).

Soil samples (10 g) were taken from the field with a 10 cm long by 5 cm diameter cylindrical tube. Surface water samples (10 mL) were collected in 50 mL sterile Falcon tubes. Three soil samples and/or three water samples, approximately 5 m apart from each other were collected from each site. All water and soil samples were stored on ice, transported to the Malaria and Dengue Laboratory (Instituto Nacional de Pesquisas da Amazônia-INPA) and processed within 24 h.

Soil samples (1 g) were suspended in 10 mL sterile water and vortexed for 10 min. Three 10-fold serial dilutions were prepared by diluting the vortexed material with sterile water. Water samples were used without any dilution. Soil and water samples were incubated at 80 °C for 12 min, for selection of sporulated *Bacillus* spp. [26,27]. Then, 50 µL aliquots were spread on Petri dishes containing nutrient agar (NA), Luria-Bertani agar (LB), or the ISP2 medium. Fluconazole (20 mg/mL) was added to the media to prevent fungal growth. All Petri dishes were incubated at 30 °C for 24 or 48 h. Negative control plates with only sterile water resulted in no colonies. The streak plate technique was applied for isolating specific bacteria from the original colonies potentially containing a mixture of microorganisms.

### 2.2. Morphological and Molecular Characterization

Colony morphology was inspected for size, shape, texture, elevation, color and Gram staining using standard microbiological techniques and a 100x magnification microscope. DNA was extracted from each colony for 16S rRNA gene amplification and sequencing as follows. DNA extraction, from isolated bacterial colonies, was performed with InstaGene™ Matrix (BioRad-Hercules, Califórnia, EUA) following the manufacturer’s instructions. DNA was spectrophotometrically quantified and adjusted to 150 ng/µL. Bacterial 16S rRNA genes were amplified by PCR using Taq Pol—Master Mix 2X (Cellco Biotec-Jardim Bandeirantes, São Carlos-SP, Brazil), and the primers 27F (5′-AGAGTTTGATCMTGGCTCAG-3′) [28] and 1100R (5′-AGGGTTGCGCTCGTT-3′) modified from [29] used in a previous study [30]. Each reaction consisted of 12.5 µL Master Mix; 1 µL DNA (150 ng/µL); 10.5 µL H_2_O milli-Q and 0.5 µL (10 pMol) of each primer. The PCR program had an initial denaturation at 95 °C for 3 min, followed by 35 cycles of (94 °C for 1 min, 54 °C for 40 s, 72 °C for 90 s), followed by a final extension at 72 °C for 5 min. Amplicon production and size were verified by electrophoresis in a 0.8% agarose gel, stained with ethidium bromide. Amplicons were purified with PCR Purification Kit (Cellco Biotec-Jardim Bandeirantes, São Carlos-SP, Brazil), following the manufacturer’s instructions and 200 ng of purified DNA was used for each sequencing reaction (BigDye Terminator V 3.1, Life Technologies-Carlsbad, Califórnia, EUA and 10 pMol of primer). The 27F and 1100R primers were used in separate sequencing reactions, generating data from both DNA strands.

All 16S rRNA gene sequences were assembled at http://asparagin.cenargen.embrapa.br/phph/ (accessed 30 May 2019) using the CAP3 program with the capability to clip 5′ and 3′ low-quality regions of reads, apply quality values in overlaps between reads, and generate consensus sequences [31]. Consensus sequences were compared with 16S sequences in GenBank applying BLASTn (https://blast.ncbi.nlm.nih.gov/Blast.cgi, accessed 30 May 2019) at the National Center for Biotechnology Information (NCBI) and Ribossomal Database Project (RDP-II http://rdp.cme.msu.edu/comparison/comp.jsp, accessed 30 May 2019). All sequences were registered in the SisGen database (Sistema Nacional de Gestão do Patrimônio Genético e do Conhecimento Tradicional Associado, A9C8D56) and are available in GenBank with accession numbers MT052595-MT052669 and MT163315-MT163316.

### 2.3. Fractionation of Bacterial Cultures

Each bacterial strain was inoculated in 2 mL of the culture medium from which it was isolated (Appendix A) and this starting culture was kept in a shaker incubator at 30 °C and 180 rpm for 24 h. Fifty µL of the initial culture was transferred to 500 mL of medium followed by incubation at 30 °C and 180 rpm for 120 h. After 120 h incubation, each culture was centrifuged at 4 °C and 2800× *g* for 40 min. The supernatant was treated as described by [18], with modifications. Following centrifugation, the supernatant was filtered through a 0.22 µM Millipore membrane and lyophilized (Terroni-Jardim Jockei Club A, São Carlos-SP) for 72 h, 150 mmHg. An aliquot of the supernatant was plated on agar-medium to certify the absence of viable cells, after centrifugation, the pellet was divided in two identical parts. One of them was stored frozen at −20 °C. The second portion was autoclaved at 127 °C for 30 min and then frozen at −20 °C. Both pellet fractions were then lyophilized as described above.

### 2.4. Mosquito Rearing

The *A. aegypti* MAO strain was maintained at 26 ± 2 °C, 80% relative humidity and 12 L: 12 D light cycle. Larvae were fed Tetramin fish food. Adults (males and females kept together) were given access to 10% sucrose solution ad libitum and were blood-fed on Hamster *Mesocricetus auratus* (W). All experiments were conducted in accordance with relevant regulations following the guidelines of the “Conselho Nacional de Controle de Experimentação Animal”—CONCEA and was approved by the “Comissao de Ética no Uso de Animais”—CEUA (053/2018-SEI 01280.001770/2018-71) of the “Instituto Nacional de Pesquisas da Amazônia”—INPA, at a meeting on 29 October 2018.

### 2.5. Screening for Larvicidal Activity

Bioassays were performed according to WHO guidelines [32], as described by Soares-da-Silva et al. [33]. Each bacterial strain was inoculated in 150 mL of the same culture medium from which it was isolated (Appendix A) and incubated at 30 °C and 180 rpm for 120 h to achieve an optical density of 8 units in the McFarland scale (~24 × 10^8^ cells/mL). Larval mortality due to exposure to bacterial culture was assessed on three consecutive days. Each day, five cups were prepared for each tested strain, each cup containing 10 third instar *A. aegypti* larvae, 9 mL of distilled water, food (Tetramin), and 1 mL of bacteria culture (bacteria plus medium). Dead larvae were counted in each cup at 24, 48 and 72 h after exposure. No mortality was observed at any time in negative controls without bacteria. *Bacillus thuringiensis israelensis* Bti-BR101 was used as positive control to validate our protocol and to comparatively assess larvicidal activity.

### 2.6. Bioassays with Fractionated Metabolites

Larvicidal activities of lyophilized supernatants and pellets were evaluated separately. Initially, the lyophilized products were prepared as follows; (i) SUP = 50 mg of supernatant + 10 mL of water; (ii) PEL = 50 mg of pellet + 10 mL of water; (iii) APEL = 50 mg of autoclaved pellet + 10 mL of water e; (iv) APEL + SUP = 25 mg of autoclaved pellet + 25 mg of supernatant + 10 mL of water. All the samples were homogenized with a vortex for 40 min. *A. aegypti* larval mortality was assessed as described above at 24, 48 and 72 h.

### 2.7. Determination of Lethal Concentrations (LC_50_) and (LC_90_)

Strains eliciting at least 50% larval mortality in previous assays were further analyzed by standardized WHO protocols [32] and lethal concentrations (LC_50_ and LC_90_) were determined separately for SUP, PEL, APEL and APEL+SUP fractions. These assays were conducted in five replicates, each containing 150 mL of water, 20 third instar larvae and 0.04, 0.03, 0.02, 0.01, 0.008, 0.005, 0.001, 0.0008 or 0.0003 mg/L of fractionated cultures. Dead larvae were counted at 24, 48 and 72 h after exposure. Negative and positive controls were included as described above using Bti BR101 as active strain. Data from concentrations causing between 10% and about 95% mortality of mosquito larvae were used for statistical analyses.

LC_50_ and LC_90_ were assessed by Probit, with *p* ≤ 0.05 [34], using the statistical software Polo Plus 1.0 (LeOra Software, Berkeley, CA, USA) [35]. Lethal concentrations and confidence interval (CI 95%) were analyzed by the Lilliefors normality test (K samples), analysis of variance (ANOVA), multiple comparison Tukey test (*p* ≤ 0.05) and Student’s t test with the software BioEstat 5.3 for Windows [36].

## 3. Results

### 3.1. Bacteria Strain Isolation and Characterization

Seventy-seven bacterial strains were isolated from soil and water samples from 15 different geographical locations (Appendix A). Among them, 38 originated from soil and 39 from water samples. Amplification and sequencing of 16S rRNA genes, and comparison of the sequences with the NCBI database, revealed that the assembled collection contains representatives of six genera: *Bacillus*, *Brevibacillus*, *Achromobacter*, *Serratia*, *Klebsiella* and *Brevundimonas*. Among them are representatives of the genus *Bacillus* and one species of each *Brevibacillus*, *Achromobacter*, *Serratia*, *Klebsiella* and *Brevundimonas* (Appendix A). Species assignment of every queried sequence would be the desired outcome, but in many cases the limited resolution of the 16S rRNA locus precluded an accurate classification at the species level. Taxonomic classification at the genus level was then assigned to the isolated strains.

### 3.2. Larvicidal Activity of Isolated Bacterial Strains

Exposure of *A. aegypti* larvae to cultured bacteria from all isolated strains (~24 × 10^8^ cells/mL) revealed that 21 resulted in a mortality equal to or greater than 50% within 72 h and these were further studied (Table 2). Twenty of them are from the genus *Bacillus*. One *Brevibacillus halotolerans* strain, SPa07, isolated from the soil killed 100% of the *A. aegypti* larvae in 48 h. Eleven of the *Bacillus* strains killed 100% of the larvae within 24 h, as did the positive control strain Bti BR101.

### 3.3. Larvicidal Activity of Fractionated Bacterial Cultures

All the 21 strains identified above were tested in another series of assays to identify larvicidal activities in cellular and secreted components separately. Bacterial cultures were submitted to centrifugation generating pellets containing bacteria and supernatant (SUP = medium/secreted molecules). Furthermore, the larvicidal activity of autoclaved pellets (APEL = unviable bacteria) was compared with the activity of untreated pellets (PEL = viable bacteria). The bacterial biomass and supernatant fractions were lyophilized before evaluating their biological activity of mosquito larvicide and determining lethal concentrations of the larvicide for 50% and 90% mortality, LC_50_ and LC_90_.

Eight out of the 21 tested strains contained one or more fractions resulting in at least 50% mortality when assayed at a concentration of 5 mg/mL (Figure 2). Larvicidal activity was observed mainly in the PEL fractions of the isolated bacteria. The PEL fractions of five strains killed 100% of the larvae in less than 24 h, as did the PEL fraction of BtiBR101 (Figure 2B). LCs determined for these five newly isolated strains were statistically indistinguishable from LCs determined for BtiBR101 (i.e., 72 h LC_90_ 0.012-0.018 mg/mL, detailed values and statistical analysis available in Table 3 and Appendix A). Autoclaved pellets lost their larvicidal activity, except for *B. safensis* SPa22, which killed 100% of *A. aegypti* larvae in 72 h (Figure 2C). APEL from SPa22 demonstrated a delayed activity with LC_50_ and LC_90_ ten-fold lower than the autoclaved pellet from the control strain BtiBR101.

SUP fractions had distinct levels of larvicidal activities, varying from *Bacillus* sp. strains SBC13 and GD02.13 with no activity at all up to 93% of mortality of *A. aegypti* larvae within 72 h in the presence of *B. safensis* SX15 SUP (Figure 2A). LCs determined for SUP fractions of SX15 and Spa07 were statistically indistinguishable from LCs determined for the positive control *B. thuringiensis* BR101 SUP (i.e., 72 h LC_90_ 0.004 mg/mL, Table 3 and Appendix A).

The mixture of SUP and APEL fractions were also tested. Noticeable results were obtained for the *Bacillus* sp. SP06 and *B. safensis* SPa22 strains. While these strains demonstrated very low larvicidal activities when SUP and APEL were tested separately, the mixture of the two fractions resulted in 100% mortality within 24 h for SP06 and 72 h for *B. safensis* SPa22. For *B. safensis* SPa22, a strain for which no larvicidal activity was detected in the SUP fraction, the APEL+SUP yielded LC_50_ and LC_90_ values lower than APEL only (Table 3). Furthermore, assays with *B. safensis* SPa22 APEL+SUP resulted in earlier lethality, 65% mortality at 24 h and 100% at 72 h (Figure 2D).

## 4. Discussion

The control of *A. aegypti* mosquitoes, vectors of diseases such as dengue, yellow fever, Zika and chikungunya, is an effective way for containing pathogen transmission to humans. However, current evidence suggests that the development of insecticide resistance is constantly challenging our ability to introduce new chemical and biological insecticides. For example, two years ago clothianidin, representing a new class of chemicals, was added by the World Health Organization to the list of insecticides indicated for indoor mosquito control. The recent report of mosquito resistance to clothianidin coming from Cameroon indicates its effectiveness for the proposed public health application may be already compromised [37].

Aiming to explore the biodiversity of the Amazon rainforest [25] and to identify novel entomopathogenic bacterial strains with larvicidal activity against *A. aegypti*, water and soil samples from 15 locations in the Amazonas state were collected and analyzed. Non-sporulated microbes were selectively eliminated by heat treatment and the surviving strains were isolated on Petri dishes containing nutritionally rich medium and agar. Seventy-seven isolated bacterial strains assembled a collection containing representatives of six genera: *Bacillus*, *Brevibacillus*, *Achromobacter*, *Serratia*, *Klebsiella* and *Brevundimonas* (Appendix A). Among them, 19 *Bacillus* sp. strains and two *Brevibacillus halotolerans* strains had larvicidal activity defined as the killing of at least 50% of the larvae within 72 h.

For the first time, strains of *Brevibacillus halotolerans* with larvicidal activity against *A. aegypti* were identified. Other bacteria of the genus *Brevibacillus*, such as *Brevibacillus laterosporus* have been characterized as having insecticidal activities against insects [38]. *Br. laterosporus* canoe-shaped spores (CSPB) contain four major proteins, CpBA, CpbB, CHRD and ExsC, that function as insect virulence factors [39]. Further studies are necessary to verify if the insecticidal activities from the *Br. halotolerans* strains isolated in this study are associated with similar proteins.

*Bacillus thuringiensis* produces, during sporulation, cellular proteinaceous crystalline toxins (Cry and Cyt proteins or d-endotoxins) with insecticidal activity against certain insect species. Additionally, vegetative *B. thuringiensis* cells secrete into the environment Vip (vegetative insecticidal proteins) and Sip (secreted insecticidal proteins), which also have insecticidal activity [40]. Commonly, the cytotoxic effect and host specificity of *B. thuringiensis*, are attributed to cellular components (Cry and Cyt) or the secreted proteins of the Vip and Sip families. However, numerous other virulence factors of *B. thuringiensis* have been discovered, including lipopeptides (surfactin, iturin, fengicin), metalloproteases, chitinases, aminopolyol antibiotics and nucleotide-mimicking moieties. These agents contribute to the insecticidal properties of *B. thuringiensis* enhancing toxin activity, avoiding host immune response evasion and participating in extracellular matrix degeneration [41,42]. Additional studies are necessary to verify if the insecticidal activities from the *Bacillus* sp. strains isolated in this study are associated with similar proteins.

Dahmana et al. [18], with the goal of evidencing and identifying multiple insecticidal components of 14 entomopathogenic bacteria against *A. albopictus* larvae, tested bacteria-free supernatant and disrupted bacterial pellet separately. Following a similar protocol, we exposed *A. aegypti* larvae to fractionated bacterial cultures; intact viable cells, autoclaved unviable cells, and bacterial culture medium depleted of cells. Larvicidal activities present in the culture medium and supernatant represent secreted molecules while cellular components are represented in the pellets. Eight out of the 21 tested strains contained one or more fractions resulting in at least 50% mortality when fractionated cultures were assayed at a concentration of 5 mg/mL. The PEL fractions of all eight strains submitted to fractionation resulted in mortality of at least 50% within 72 h, five of them killing 100% of the *A. aegypti* larvae in less than 24 h. LC_50_ and LC_90_ determined for these strains are similar to those of the reference *B. thuringiensis* strain BR101, suggesting they could be developed commercially as *A. aegypti* control agents. Autoclaved pellets lost most of their larvicidal activity, likely due to denaturation of the active components and/or the destruction of mechanisms requiring cell metabolism and replication. Notably, the autoclaved pellet from *B. safensis* SPa22 demonstrated consistently a delayed effect, killing *A. aegypti* larvae only after 48 h of exposure (Figure 2C). The mechanism involved in this late action deserves investigation.

Mortality of *A. aegypti* larvae was detected after exposure to SUP fractions of four strains, indicative that these bacteria secrete molecules with larvicidal activities. Possibly, *B. safensis* found in our study is similar to that described by [43], with culture supernatant effective against *A. aegypti*. Further investigation is necessary to uncover their molecular composition and how these agents contribute to the observed insecticidal properties.

We also tested mixing APEL and SUP fractions to evaluate the possibility of synergistic interactions between secreted compounds and cellular constituents. Synergism between *B. thuringiensis* cellular and secreted factors of selected strains have already been described [42,44]. The mixture APEL+SUP from *Bacillus* sp. SP06 and *B. safensis* SPa22 strains resulted in larvicidal activity superior to the sum of those of each fraction separately indicative of synergism between the components of APEL and SUP fractions.

## 5. Conclusions

We isolated Amazonian bacterial strains that exhibit larvicidal activity comparable to a commercial bacterial insecticide product (Bti) against *A. aegypti*. The data are promising for potentially developing novel bioinsecticides for the control of mosquitoes of medical importance. Larvicidal activities of their separated supernatant and pellet fractions were investigated and studies are needed to characterize their insecticidal compounds and mechanisms of action.

## Figures and Tables

**Figure 1 tropicalmed-06-00104-f001:**
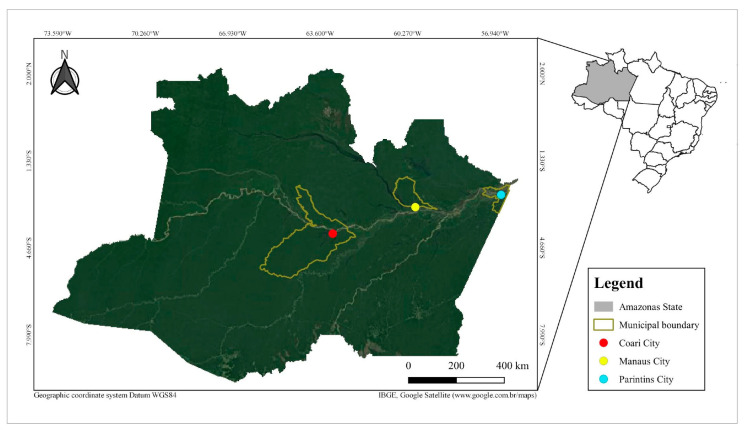
Sample collection locations. GPS coordinates, dates of collections and descriptions of collection sites are available in Table 1.

**Figure 2 tropicalmed-06-00104-f002:**
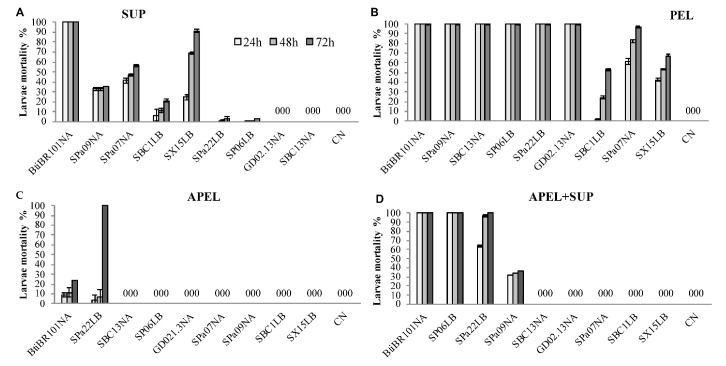
Larvicidal activity of microbial cultures fractions at 5 mg/mL. (**A**) SUP = 50 mg of lyophilized supernatant + 10 mL of water; (**B**) PEL = 50 mg of lyophilized pellet + 10 mL of water; (**C**) APEL = 50 mg of autoclaved and lyophilized pellet + 10 mL of water and; (**D**) APEL+SUP = 25 mg of autoclaved and lyophilized pellet + 25 mg of lyophilized supernatant + 10 mL of water. *A. aegypti* larvae mortality was evaluated at 24, 48 and 72 h and the results are expressed as averages ± standard deviations. CN—negative control with water in place of microbial culture fractions.

**Table 1 tropicalmed-06-00104-t001:** Geographic location, date and characteristics of the 15 sites where water and soil were collected.

Municipality	Collection Site		GPS Coordinates	Description	Sample Type	Date
Coari	1	Sítio do Gordo	4°06′45.5″ S 63°07′44.0″ W	artificial lake	water	02/2017
Manaus	2	Ramal do 7 (Brasileirinho)	3°01′42.2″ S 59°52′15.6″ W	primary forest	soil	03/2017
	3	Campus-UFAM	3°05′48.8″ S 59°58′23.6″ W	residual forest	soil	04/2017
	4	Bosque da Ciência-INPA	3°06′01.3″ S 59°59′06.6″ W	residual forest	soil	02/2017
	5	Casa 15-INPA	3°05′47.2″ S 59°59′09.5″ W	residual forest	soil	05/2017
	6	Sitio Portela 1	3°02′47.0″ S 59°52′54.4″ W	fish tank	water	02/2017
	7	Sitio Dona Chagas	3°02′33.5″ S 59°53′15.6″ W	fish tank	water	06/2017
	8	Sítio Portela 2	3°02′47.5″ S 59°52′56.8″ W	fish tank	water	02/2017
Parintins	9	Parananema	2°40′30.9″ S 56°45′59.2″ W	primary forest	soil	03/2017
	10	Aninga	2°39′07.6″ S 56°46′50.3″ W	natural pond	water	03/2017
	11	Lagoa Francesa	2°37′34.7″ S 56°43′37.3″ W	natural pond	water	04/2017
	12	Areial	2°39′39.2″ S 56°46′07.6″ W	creek	water	03/2017
	13	Macurany	2°40′33.6″ S 56°45′30.2″ W	primary forest	soil	04/2017
	14	Parananema	2°40′30.9″ S 56°45′59.2″ W	creek	water	02/2017
	15	Macurany (Sítio Fanuel)	2°39′06.7″ S 56°43′29.0″ W	natural pond	water	04/2017

**Table 2 tropicalmed-06-00104-t002:** Isolated strains with larvicidal activity against *A. aegypti* larvae.

Strain	Isolated from	GenBank acc. N°.; Species	Cumulative Mortality		
			24 h	48 h	72 h
BR101	Positive control	*Bacillus thuringiensis var.israelensis*	100%	-	-
Spa09	Soil	MT052636; *Bacillus* sp.	100%	-	-
SBC13	Soil	MT052634; *Bacillus* sp.	100%	-	-
SP06	Soil	MT052669; *Bacillus* sp.	100%	-	-
SPO2	Water	MT052609; *Bacillus safensis*	100%	-	-
SPO5	Water	MT052624; *Bacillus* sp.	100%		
SX02	Water	MT052611; *Bacillus* sp.	100%		
APR6I	Water	MT052596; *Bacillus* sp.	100%	-	-
APR10I	Water	MT052598; *Bacillus* sp.	100%	-	-
GD02.13	Water	MT163315; *Bacillus* sp.	100%	-	-
SX06	Water	MT052643; *Bacillus megaterium*	100%		
SX08	Water	MT052649; *Bacillus velezensis*	100%		
Spa07	Soil	MT052647; *Brevibacillus halotolerans*	60%	100%	-
Spa03	Soil	MT052618; *Bacillus* sp.	90%	97%	100%
LFP2	Water	MT052629; *Bacillus subtilis*	10%	80%	90%
Spa22	Soil	MT052639; *Bacillus safensis*	47%	80%	83%
Spa04	Soil	MT052633; *Brevibacillus halotolerans*	70%	70%	80%
SMP1.2	Soil	MT052614; *Bacillus* sp.	56%	63%	67%
SBC2	Soil	MT052620; *Bacillus subtilis*	27%	57%	67%
SBC1	Soil	MT052667; *Bacillus* sp.	33%	53%	63%
Spa14	Soil	MT052651; *Bacillus* sp.	37%	57%	57%
SX15	Water	MT163316; *Bacillus safensis*	-	67%	67%

Mortality was assessed at 24, 48 and 72 h after exposure to bacterial cultures diluted 10-fold in water. No mortality was observed in negative controls without bacteria and the strain BR101 was used as positive control, inducing 100% mortality within 24 h. Values are the average of three biological replicates, each challenging 10 third instar larvae in 9 mL of distilled water and 1 mL (~24 × 10^8^ cells/mL) of bacterial culture (bacteria plus medium).

**Table 3 tropicalmed-06-00104-t003:** LC_50_ and LC_90_ values of microbial culture fractions against *A. aegypti* larvae.

Strain	PEL						Strain	APEL					
	24 h		48 h		72 h			24 h		48 h		72 h	
	CL_50_	CL_90_	CL_50_	CL_90_	CL_50_	CL_90_		CL_50_	CL_90_	CL_50_	CL_90_	CL_50_	CL_90_
Bti	(0.008)bc	(0.058)a	(0.008)a	(0.023)a	(0.003)a	(0.012)a	Bti	-	-	-	-	-	-
SPa09	(0.006)c	(0.089)a	(0.006)a	(0.026)a	(0.005)a	(0.015)a	SPa09	-	-	-	-	-	-
SPa22	(0.013)ab	(0.113)a	(0.004)ab	(0.023)a	(0.005)a	(0.015)a	SPa22	(0.009)a	(0.073)a	(0.006)a	(0.025)a	(0.007)a	(0.018)a
GD02.13	(0.006)c	(0.148)a	(0.005)ab	(0.017)a	(0.003)a	(0.012)a	GD02.13	-	-	-	-	-	-
SP06	(0.009)bc	(0.083)a	(0.007)ab	(0.029)a	(0.007)a	(0.018)a	SP06	-	-	-	-	-	-
SBC13	(0.007)c	(0.009)a	(0.004)b	(0.019)a	(0.005)a	(0.012)a	SBC13	-	-	-	-	-	-
SPa07	(0.009)bc	(0.124)a	(0.005)ab	(0.021)a	-	-	SPa07	-	-	-	-	-	-
SX15	(0.010)bc	(0.088)a	(0.005)ab	(0.022)a	-	-	SX15	-	-	-	-	-	-
BC1	-	-	-	-	(0.023)a	(0.256)a	BC1	-	-	-	-	-	-
**Strain**	**SUP**						**Strain**	**APEL+SUP**					
	**24 h**		**48 h**		**72 h**			**24 h**		**48 h**		**72 h**	
	CL_50_	CL_90_	CL_50_	CL_90_	CL_50_	CL_90_		CL_50_	CL_90_	CL_50_	CL_90_	CL_50_	CL_90_
Bti	(0.009)a	(0.048)a	(0.008)a	(0.029)a	(0.004)a	(0.004)a	Bti	(0.005)a	(0.037)a	(0.005)a	(0.018)a	(0.004)a	(0.011)a
SPa09	-	-	-	-	-	-	SPa09	-	-	-	-	-	-
SPa22	-	-	-	-	-	-	SPa22	(0.007)a	(0.031)a	(0.007)a	(0.018)a	(0.003)a	(0.012)a
GD02.13	-	-	-	-	-	-	GD02.13	-	-	-	-	-	-
SP06	-	-	-	-	-	-	SP06	(0.008)a	(0.046)a	(0.006)a	(0.023)a	(0.003)a	(0.016)a
SBC13	-	-	-	-	-	-	SBC13	-	-	-	-	-	-
SPa07	(0.007)a	(0.056)a	(0.006)a	(0.020)a	(0.004)a	(0.004)a	SPa07	-	-	-	-	-	-
SX15	(0.006)a	(0.068)a	(0.006)a	(0.068)a	(0.005)a	(0.005)a	SX15	-	-	-	-	-	-
BC1	-	-	-	-	-	-	BC1	-	-	-	-	-	-

Assays were conducted in five replicates, each containing 150 mL of water, 20 third instar larvae and 0.04, 0.03, 0.02, 0.01, or 0.005 mg/L of fractionated cultures. Dead larvae were counted at 24, 48 and 72 h after exposure to the samples. LC50 and LC90 were assessed by Probit, with *p* ≤ 0.05. Statistical comparisons and confidence inter-val (CI 95%) were analyzed by the Lilliefors normality test (K samples), analysis of variance (ANOVA), multiple comparison Tukey test (*p* ≤ 0.05) and Student’s t test and the results are presented in the Appendix A. For all variables in each column (a.b.c) with the same letter, the difference between the values is not statistically significant.

## Data Availability

All data generated or analyzed during this study are included in this published article and its Appendix A. DNA sequences are available in the database GenBank at National Center for Biotechnology Information www.ncbi.nlm.nih.gov accessed on 14 June 2021, with access numbers MT052595-MT052669 and MT163315-MT163316.

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
