# Peer review of "Larvicidal Activities against Aedes aegypti of Supernatant and Pellet Fractions from Cultured Bacillus spp. Isolated from Amazonian Microenvironments"

_tropicalmed, 2021, doi:10.3390/tropicalmed6020104_

Round 1

Reviewer 1 Report

In this manuscript, Katak et al., explore the larvicidal potential of cultivable environmental bacteria isolated from soil and water samples in the Amazonian region against Aedes aegypti. The authors report 77 bacterial strains isolated, most related to the genus Bacillus, in which around 30% presented the larvicidal activity to a certain extent. The work is presented in a clean and easy-to-read manner, and I hope that the simplicity of the results reflects the initial steps towards a more mechanistic exploration of the strains here identified, in studies to follow. That said, I do have a single major comment that I would like to be addressed, along with a few minor suggestions and corrections that I believe would improve the manuscript, prior to publication. 

Minor comments:

There are quite a few misspelled words/typos and inconsistencies that I would recommend the authors fix. I will not spend their time going over every single one of them, but will rather show three examples:

L29: "entompatogenic" and "This study"

L61: "zika" and in other parts "Zika"

L159: "testes"

L355: Brevacillus halotolerans is not italicized. 

L67-71: I would recommend reducing the first paragraph to a few sentences and increase the size of this paragraph, to provide better context and examples of entomopathogenic organisms and strategies currently being used for vector control.

 L83: Could the authors include the season in which samples were collected? As microbial community shows seasonal variation.

Table 1: Correct the spacing in the first and last columns, the word "Municipality" and "2017" are getting cut 

Table 1: In an effort to improve data quality, I'd recommend that in the future, the authors include climatic and environmental data such as temperature, humidity, atmospheric pressure, etc, for their collection sites.

L103: How many samples were collected per collection site?

L108: Was the storage method for transportation similar for soil and water samples?  As it is written, It seems that only the water sample storage process is specified.

L137-139: Did the authors at some point cross-examined their consensus sequences with other well-curated databases such as the SILVA?

L153: Here and throughout the manuscript, I would suggest replacing the word insecticidal with larvicidal, given the nature of the assays employed in which only larvae were tested.

Table 2: Please specify it as cumulative mortality

Figure 2: Panel C has an issue with the label on the Y-axis, and please, define CN as part of your figure legend.

Discussion:

Would it be possible to combine the results with the discussion? As it currently stands, there is not much of a discussion of the results obtained in this study, but rather a repetition of what is already presented in the results section.

L343: Selection "for" resistant insects. Additionally, are there any reports indicating that this might be the case? Chemical resistance is dramatically different from resistance to a biological agent. 

L358-368: I would recommend moving this to the introduction, as Bti is not the focus of this study, but rather sets up the rationale behind this work.

Major comment:

L115: The manuscript currently lacks key information on appropriate controls used for their sequencing effort. For instance, were any blank controls (no sample or source of genetic template like sterile water and sequencing kit reagent alone) used along with samples of interest at each sample processing step? Namely: sample collection, DNA extraction, and purification, PCR amplification? Should also inform if sample processing was performed by the same individual(s). Were any mock controls with spiked microbial communities ran in parallel?

Author Response

Response to Reviewer 1 Comments

Please find below our responses, in bold and underlined font, to the reviewers’ comments and suggestions.

Reviewer 1

In this manuscript, Katak et al., explore the larvicidal potential of cultivable environmental bacteria isolated from soil and water samples in the Amazonian region against Aedes aegypti. The authors report 77 bacterial strains isolated, most related to the genus Bacillus, in which around 30% presented the larvicidal activity to a certain extent. The work is presented in a clean and easy-to-read manner, and I hope that the simplicity of the results reflects the initial steps towards a more mechanistic exploration of the strains here identified, in studies to follow. That said, I do have a single major comment that I would like to be addressed, along with a few minor suggestions and corrections that I believe would improve the manuscript, prior to publication. 

Minor comments:

There are quite a few misspelled words/typos and inconsistencies that I would recommend the authors fix. I will not spend their time going over every single one of them, but will rather show three examples:

The manuscript was carefully inspected, and spelling mistakes were fixed

L29: "entompatogenic" and "This study" corrected

L61: "zika" and in other parts "Zika" corrected

L159: "testes" corrected

L355: Brevacillus halotolerans is not italicized.  corrected

L67-71: I would recommend reducing the first paragraph to a few sentences and increase the size of this paragraph, to provide better context and examples of entomopathogenic organisms and strategies currently being used for vector control. We expanded the text adding examples of entomopathogenic organisms

 L83: Could the authors include the season in which samples were collected? As microbial community shows seasonal variation. In the Amazon forest, there are no periodic seasons such as summer, winter, autumn, and spring by virtue of the tropics. We decided to inform only the dates of collections.

Table 1: Correct the spacing in the first and last columns, the word "Municipality" and "2017" are getting cut  corrected

Table 1: In an effort to improve data quality, I'd recommend that in the future, the authors include climatic and environmental data such as temperature, humidity, atmospheric pressure, etc, for their collection sites. We will do it in future studies

L103: How many samples were collected per collection site? Text added: Three soil samples and/or three water samples, approximately 5 meters apart from each other. were collected in each site,

L108: Was the storage method for transportation similar for soil and water samples?  As it is written, It seems that only the water sample storage process is specified. corrected

L137-139: Did the authors at some point cross-examined their consensus sequences with other well-curated databases such as the SILVA? No, but we used two other comprehensive databases, NCBI and RDP-II, with consistent results.  

L153: Here and throughout the manuscript, I would suggest replacing the word insecticidal with larvicidal, given the nature of the assays employed in which only larvae were tested. corrected

Table 2: Please specify it as cumulative mortality corrected

Figure 2: Panel C has an issue with the label on the Y-axis, We do not see a problem in the figure 2C,  and please, define CN as part of your figure legend. corrected

Discussion:

Would it be possible to combine the results with the discussion? As it currently stands, there is not much of a discussion of the results obtained in this study, but rather a repetition of what is already presented in the results section. The Discussion text was shortened avoiding the repletion of information from the Results section

L343: Selection "for" resistant insects. Additionally, are there any reports indicating that this might be the case? Chemical resistance is dramatically different from resistance to a biological agent.  We could not understand what the reviewer is asking for.

L358-368: I would recommend moving this to the introduction, as Bti is not the focus of this study, but rather sets up the rationale behind this work. corrected

Major comment:

L115: The manuscript currently lacks key information on appropriate controls used for their sequencing effort. For instance, were any blank controls (no sample or source of genetic template like sterile water and sequencing kit reagent alone) used along with samples of interest at each sample processing step? Namely: sample collection, DNA extraction, and purification, PCR amplification? Should also inform if sample processing was performed by the same individual(s). Were any mock controls with spiked microbial communities ran in parallel? A sentence was added to materials and methods indicating our controls for contamination during bacteria isolation. DNA sequencing was performed with DNA extracted from isolate colonies, reducing or eliminating the possibility of confounding minor contaminations in sequencing reagents.

Reviewer 2 Report

In this study, Katak et al. collected and isolated Bacillus bacteria species from Amazonian field. The mosquito larva bioassay was conducted to identify the toxicity of the purified bacteria to Aedes aegypti. The purpose of this study is to select effective larvicidal Bacillus species as an alternative strategy for mosquito control in future. Authors employed appropriate methods, and the writing of this manuscript is fluent. Here are a few comments for authors to modify the manuscript.

Major revise:

  1. Need to provide more detailed description in the material and methods. 
  2. The bioassay concentrations of bacteria species need to cause 10~95% mortality of mosquito larvae. Authors should mention it if it was.
  3. The food shouldn’t be provided to mosquitoes in first 24h exposure period. Please add more detail about food feeding during the bioassay.
  4. The “Fractionation of bacterial cultures” could be present prior to “Screening for Insecticidal activity” to make the method fluent.
  5. In data analysis (Figure 2), it would be better to determine the significance of the mortality rates among 24, 48, 72hr since the standard deviations included.

Minor revise:

It should be “5 mg/ml” in abstract.

Author Response

Response to Reviewer 2 Comments

Please find below our responses, in bold and underlined font, to the reviewers’ comments and suggestions.

Reviewer 2

In this study, Katak et al. collected and isolated Bacillus bacteria species from Amazonian field. The mosquito larva bioassay was conducted to identify the toxicity of the purified bacteria to Aedes aegypti. The purpose of this study is to select effective larvicidal Bacillus species as an alternative strategy for mosquito control in future. Authors employed appropriate methods, and the writing of this manuscript is fluent. Here are a few comments for authors to modify the manuscript.

Major revise:

  1. Need to provide more detailed description in the material and methods. Additional information was added to the materials and methods section of the manuscript

  1. The bioassay concentrations of bacteria species need to cause 10~95% mortality of mosquito larvae. Authors should mention it if it was. Information was added to the manuscript

  1. The food shouldn’t be provided to mosquitoes in first 24h exposure period. Please add more detail about food feeding during the bioassay. According to WHO guidelines cited in our manuscript “For  long  exposures,  larval  food  should  be  added  to  each  test  cup,  particularly  if  high  mortality  is  noted  in  control.”

  1. The “Fractionation of bacterial cultures” could be present prior to “Screening for Insecticidal activity” to make the method fluent. corrected

  1. In data analysis (Figure 2), it would be better to determine the significance of the mortality rates among 24, 48, 72hr since the standard deviations included. We consider that adding letters indicating significance will make the graphics too busy and the information is not relevant. The objective of experiments displayed in table 2 and Figure 2 is to identify and select strains and bacterial fractions with noteworthy larvicidal activities. Statistical significances are presented in Table 3, comparing the isolated strains with the positive control Bti.

Minor revise:

  1. It should be “5 mg/ml” in abstract. corrected

Round 2

Reviewer 2 Report

Authors have responded comments and revised the manuscript.